# Is Acrylamide as Harmful as We Think? A New Look at the Impact of Acrylamide on the Viability of Beneficial Intestinal Bacteria of the Genus *Lactobacillus*

**DOI:** 10.3390/nu12041157

**Published:** 2020-04-21

**Authors:** Katarzyna Petka, Tomasz Tarko, Aleksandra Duda-Chodak

**Affiliations:** 1Department of Plant Products Technology and Nutrition Hygiene, Faculty of Food Technology, University of Agriculture in Krakow, 30-149 Krakow, Poland; katarzyna.petka@urk.krakow.pl; 2Department of Fermentation Technology and Microbiology, Faculty of Food Technology, University of Agriculture in Krakow, 30-149 Krakow, Poland; tomasz.tarko@urk.edu.pl

**Keywords:** lactic acid bacteria, probiotic, acrylamide, viability, flow cytometry

## Abstract

The impact of acrylamide (AA) on microorganisms is still not clearly understood as AA has not induced mutations in bacteria, but its epoxide analog has been reported to be mutagenic in *Salmonella* strains. The aim of the study was to evaluate whether AA could influence the growth and viability of beneficial intestinal bacteria. The impact of AA at concentrations of 0–100 µg/mL on lactic acid bacteria (LAB) was examined. Bacterial growth was evaluated by the culture method, while the percentage of alive, injured, and dead bacteria was assessed by flow cytometry after 24 h and 48 h of incubation. We demonstrated that acrylamide could influence the viability of the LAB, but its impact depended on both the AA concentration and the bacterial species. The viability of probiotic strain *Lactobacillus acidophilus* LA-5 increased while that of *Lactobacillus plantarum* decreased; *Lactobacillus brevis* was less sensitive. Moreover, AA influenced the morphology of *L. plantarum,* probably by blocking cell separation during division. We concluded that acrylamide present in food could modulate the viability of LAB and, therefore, could influence their activity in food products or, after colonization, in the human intestine.

## 1. Introduction

Acrylamide (AA) is a chemical compound used in many industries. It is produced as a substrate for the synthesis of polymers widely used in the paper, chemical, and cosmetics industries. In 1994, the International Agency for Research on Cancer (IARC) included acrylamide in a group of compounds “probably carcinogenic to humans” after laboratory tests in mice and rats [1]. 

Acrylamide in foods is formed mainly by the reaction of free asparagine with reducing sugars (especially fructose and glucose) during the Maillard reaction, but it can also be formed by other pathways, e.g., the acrolein pathway [2]. The most important factors for AA formation are time and the temperature of the thermal processing of food products, and it is thought that a prerequisite for AA formation is temperature exceeding 120 °C. 

Acrylamide has been shown to be a reproductive toxicant in animal models [3,4]. It exerts neurotoxic activity [5,6,7], and many studies have proved that AA also has genotoxic, cytotoxic, and carcinogenic impacts on the human organism [6,8,9,10]. However, due to the fact that acrylamide does not exert a mutagenic effect in bacterial cells [3,11], it has been agreed that its carcinogenic activity is related to glycidamide (GA)—an acrylamide metabolite formed in mammalian cells. The mutagenic and genotoxic effects of GA have already been confirmed in various in vitro and in vivo studies, showing that this AA metabolite can induce the formation of DNA adducts, resulting in mutagenesis and the development of cancers [6,8,9,12]. 

The impact of AA on microorganisms is still unclear. The results of many assays made by various laboratories are consistent in showing that AA is not a mutagen in *Salmonella* Typhimurium tested strains at concentrations up to 5 mg/plate, with or without metabolic activation [3]. However, three epoxide analogs of acrylamide, e.g., glycidamide, have been reported to be mutagenic in *Salmonella* strains ± S9 activation [11,13]. Tsuda et al. [14] reported that AA did not induce any gene mutations in *Salmonella*/microsome test systems (TA98, TA100, TA1535, TA1537) and in *Escherichia coli*/microsome assays (WP2 uvrA^−^) up to a dose of 50 mg AA/plate, but acrylamide did show a strong positive response in a *Bacillus subtilis* spore-rec assay (induced DNA damage) at 10–50 mg/disc. According to the authors, the results suggested that AA had the potential to induce gross DNA damage rather than point mutations detected by the Ames test. There are also studies demonstrating that after introducing 1%–3% acrylamide into the growth medium, *Escherichia coli* cells undergo various changes, such as blockage of cell division, elongation of cells, inhibition of DNA synthesis, decreased osmotic stability, and ultrastructural alterations of the outer membrane [15].

Taking into account eukaryotic cells, it is worth citing the research of Kwolek-Mirek et al. [16]. They demonstrated that acrylamide caused impairment of growth of *Saccharomyces cerevisiae* yeast deficient in Cu, Zn-superoxide dismutase (Δsod1) in a concentration-dependent manner. This inhibitory effect was not due to cell death but to decreased cell vitality and proliferative capacity. Exposing Δsod1 yeast to acrylamide caused the increased generation of reactive oxygen species and decreased glutathione levels.

It has also been proven that some microorganisms have the ability to use acrylamide as a carbon and nitrogen source for their growth and that amidases are the main factor involved in AA degradation. Amidases are enzymes (EC. 3.5.1.4) that occur ubiquitously in nature and are characterized by a broad spectrum of catalyzed reactions [17]. Classification on the basis of catalytic activity takes into account the substrate specificity profile of the particular amidase and divides known amidases into six classes. During the amidase-catalyzed deamination reaction of acrylamide, acrylic acid and ammonia are formed. Then, acrylic acid can be reduced to propionate or transformed into β-hydroxypropionate, lactate, or CO_2_, in a pathway involving coenzyme-A [2,5,8]. 

To date, laboratory tests have shown the ability to degrade AA by many environmental microorganisms, mainly bacteria, such as *Ralstonia eutropha* [18], *Pseudomonas chlororaphis* [19], *Enterobacter aerogenes* [20], *Pseudomonas aeruginosa* [21,22], *Bacillus cereus* [23], *Rhodococcus* sp., *Klebsiella pneumoniae* [24,25], and *Burkholderia* sp. [26]. It is worth highlighting that among the amidase producers are certain species that naturally occur in human organisms or are delivered with food, such as *Escherichia coli* [27], *Bacillus clausii* [28], *Enterococcus faecalis* [29], and *Helicobacter pylori* [30,31]. However, the substrate specificity of their amidases and the potential for reaction with acrylamide have not yet been confirmed. In some cases, it has even been proved that those bacteria produce only cell wall amidases, such as N-acetylmuramoyl-L-alanine amidase [32], with no affinity to acrylamide. Either way, there is a possibility that members of microbiota could degrade acrylamide directly in the human intestine. 

Lactic acid bacteria (LAB) constitute very important members of intestinal microbiota and play an important role in proper organism functioning and maintenance of our health [33,34,35,36]. Representatives of LAB are also important in the food industry, both as starter culture added during production and as native microbiota of raw materials used for food production [37,38,39]. The positive role of LAB could also be related to their ability to reduce AA levels in organisms or foodstuffs. To date, the possibility of degrading AA by amidase production has not been confirmed, although synthesis of N-acetylmuramoyl-L-alanine amidase, involved in the degradation of peptidoglycan and hydrolysis of the amide bond between N-acetylmuramic acid and L-amino acids of the bacterial cell wall, has been reported in LAB [40,41]. Other studies [42,43] have shown that *Lactobacillus reuteri* NRRL 14171 and *Lactobacillus casei* Shirota are able to remove acrylamide in aqueous solution by physically binding the toxin to the bacterial cell wall, probably with a significant role of the teichoic acid structure. Later, Rivas-Jimenez [44] demonstrated that both mentioned bacterial strains were able to remove dietary AA (commercial potato chips with an average AA content of ~34,000 µg/kg) under different simulated gastrointestinal conditions. The percentage of AA removed by each bacterium exposed to different concentrations of the toxin (10–350 µg/mL) had a similar tendency; the lower the concentration of AA, the higher the percentage of toxin removed. The results showed that *L. casei* Shirota showed a higher percentage (68%) of AA removed than *L. reuteri* (53%) when bacteria were exposed to the lowest concentration of toxin (10 µg/mL), but no significant differences (*p* < 0.05) were observed in the percentage of toxin removed by both strains (~2%) when ≥100 mg/mL of AA was used. These findings proved that strains of the genus *Lactobacillus* could be employed to reduce the bioavailability of dietary AA. However, the strong dependence on AA concentration suggests that the mechanism of AA reduction is still the physical binding of AA by bacteria. 

To the best of our knowledge, no one has investigated how acrylamide affects the viability of lactic acid bacteria so far, and this is an important issue considering their important role in the human body. First of all, lactic acid bacteria can be exposed to acrylamide just in food products. There are many fermented milk products that contain various “additives” rich in AA, such as biscuits, muesli, roasted almonds, nuts and seeds, dried fruit, breakfast cereals, and bran flake cereals. Also, so-called pro-health foods, such as probiotic bars and cereals, contain live strains of LAB, as well as crispy cereals, roasted nuts, almonds and seeds, almond and peanut butter, dried fruits, flakes, etc. Moreover, intestinal LAB can also be exposed to dietary acrylamide after intake of various fried, grilled, toasted, roasted, or baked foods. Although acrylamide is rapidly absorbed from the intestine, there are studies suggesting that some food matrices (or components) can reduce the intestinal absorption of AA. For example, a high protein concentration in the human diet may reduce acrylamide uptake [45], causing unmetabolized acrylamide to reach the colon. Therefore, the aim of this study was to evaluate whether acrylamide could influence the growth and viability of lactic acid bacteria belonging to the *Lactobacillus* genus.

## 2. Materials and Methods

### 2.1. Bacteria

Pure cultures of lactic acid bacteria belonging to the *Lactobacillus* genus were used in the study. For the experiments, 4 strains constituting a typical microbiota of fermented milk products and 2 probiotic strains were chosen: *Lactobacillus plantarum* DSMZ 20205, *Lactobacillus brevis* DSMZ 20054, *Lactobacillus lactis* subsp. *lactis* DSMZ 20481, and *Lactobacillus casei* DSMZ 20011. All were purchased from Leibniz Institut DSMZ (Deutsche Sammlung von Mikroorganismen und Zelkulturen GmbH, Braunschweig, Germany). Two probiotic strains—*Lactobacillus acidophilus* LA-5 and *L. casei* LC01—were obtained from Christian Hansen (Hørsolm, Denmark). 

Bacteria were delivered as freeze-dried cultures and were handled according to supplier protocol. Briefly, after opening the ampoule, bacteria were rehydrated and then transferred to a tube with sterile liquid De Man, Rogosa, and Sharpe (MRS) agar medium (BioMaxima, Lublin, Poland) and incubated at a temperature optimal for strain. For *L. acidophilus* LA-5 and both *L. casei* strains, the optimal temperature was 37 °C, while, for other *Lactobacillus* species, it was 30 °C. 

### 2.2. Measurement of Optical Density of Bacterial Suspension: Calibration

To tubes containing 5 mL of sterile MRS medium, a volume of 0.1 mL of 24-h liquid bacterial culture was added, the contents were mixed, and the tubes were incubated for 24 h at the optimum temperature for the tested strain. After incubation, bacterial cultures were centrifuged at 194× *g* for 15 min (MPW-35JR centrifuge, MPW MED Instruments, Warsaw, Poland), and the supernatant was discarded. The pellets were rinsed by mixing with 5 mL of sterile distilled water followed by centrifugation (using previous parameters). The resulting pellets were resuspended in sterile water so as to obtain an optical density of the bacterial suspensions equal to McFarland standard 1.0 (using a Den-1B densitometer, Biosan, Latvia). Then, serial 10-fold dilutions were made in sterile water, and 1 mL of subsequent dilution was spread over the surface of the MRS medium (in triplicate). After 72 h of incubation at an optimal temperature, bacterial colonies were counted, mean bacterial cell density in cfu/mL from 3 replicates was calculated for each tested strain, and the relationship between the optical density of McFarland = 1 and bacterial cell density was determined. The relationships obtained for individual strains were as follows (1 McFarland unit equivalent): *L. plantarum*, 1.55 × 10^8^ cfu/mL; *L. brevis*, 4.5 × 10^7^ cfu/mL; *L. lactis* subsp. *lactis*, 1.6 × 10^8^ cfu/mL: *L. casei*, 4.9 × 10^7^ cfu/mL; *L. acidophilus* LA-5, 4.45 × 10^7^ cfu/mL; *L. casei* LC01, 4.8 × 10^7^ cfu/mL. Before each experiment, a 24 h culture of adequate *Lactobacillus* strain was centrifuged, washed in sterile water, and resuspended (as described above). The optical density of the bacterial suspension was adjusted to a value corresponding to 2 × 10^7^ cfu/mL.

### 2.3. Model Medium for Experiments

All experiments were carried out in carbon- and nitrogen-limiting conditions because model medium composed of 0.45% NaCl (POCh, Gliwice, Poland), and 0.45% bacteriological peptone (BioMaxima, Lublin, Poland) was used. If a solid medium was required, bacteriological agar was added in a final concentration of 2% (BioMaxima, Lublin, Poland). All media were sterilized using a Microjet Microwave Autoclave (process parameters: 135 °C, 80 s, 3.6 bar; Enbio Technology Sp. z o.o., Gdynia, Poland). 

### 2.4. Preparation of Acrylamide “Stock” Solution

Concentrated (20 g/L) aqueous solution of acrylamide (purum, ≥98% (GC) provided by Sigma-Aldrich Sp. z o.o, Poznan, Poland) was sterilized by filtering through a sterile membrane filter (pore φ = 0.22 µm; PES Millex-GP, Bionovo, Poland) and diluted (if needed) with sterile distilled water to obtain “stock” solutions of acrylamide (concentrations: 0.5, 1.0, 2.0, 5.0, 10.0, and 20.0 g/L). 

### 2.5. Preliminary Assessment of Acrylamide Impact on Lactobacillus Growth

The impact of acrylamide on *Lactobacillus* was assessed by evaluating visible bacterial growth on the solid model medium containing acrylamide at various concentrations: 10, 50, 100, 250, 500, and 1000 µg/mL. Serial 10-fold dilutions of the suspension of tested bacteria (2 × 10^7^ cfu/mL) were made in sterile water. Then, a volume of 1 mL of acrylamide “stock” solution of adequate concentration was added to 18 mL of sterile, cooled, but still, liquid, model medium and poured into a sterile Petri plate containing 1 mL of the diluted bacterial suspension. Positive controls were Petri plates with 19 mL of the model medium (without acrylamide) mixed with 1 mL of a diluted suspension of tested bacteria. After media solidification, all plates were incubated for 72 h at a proper temperature optimal for the tested strain, and then the bacterial growth was assessed according to the following scale:
++++ very intense growth (colonies cover the whole surface, creating lawn plates)+++ intense growth (too many colonies to count, but they are distinguishable)++ good growth (30–300 colonies/plate)+ only a few colonies (<30 colonies/plate)− no growth

First, the growth of bacteria on plates with positive control was evaluated, and the dilution of bacterial suspension with good growth (30–300 colonies/plate) was chosen. For the same dilution, growth in the presence of AA was assessed. The experiment was performed in 5 replicates.

### 2.6. Determination of Cell Concentration and Viability by Flow Cytometry

The *Lactobacillus* strains whose growth was influenced by acrylamide in the preliminary analysis were chosen for this stage of the experiment. A volume of 1 mL of bacterial suspension (containing 2 × 10^7^ cfu/mL) was inoculated into 19 mL of liquid model medium, with the addition of acrylamide to a final concentration of 7.5, 15, 30, or 100 µg/mL, and incubated for 48 h. The final bacterial cell density was 10^6^ cells/mL, which corresponded to the average number of LAB cells found in fermented milk drinks (FAO/WHO Food Standards). The proposed AA concentrations were selected based on the literature [42,46], and the 100 µg/mL concentration is higher than the possible level reached in the human gastrointestinal tract or in food products. The positive control was medium with 1 mL of sterile distilled water added instead of an acrylamide “stock” solution (marked as 0 µg/mL). Immediately after adding bacteria to the medium (marked as 0 h, but taking into account staining times and cytometric measurement, the analysis was actually done about 2 h after adding the bacteria), after 24 h and 48 h of incubation at an optimal temperature, the cell concentration (cell/mL) was evaluated by flow cytometry (BD Accuri^TM^ C6 Flow cytometer, BD Biosciences, Bio-Rad, Poland) equipped with fluorescence detectors FL1 533/30, FL2 585/40, FL3 670LP. For this purpose, the commercially available BD™ Cell Viability Kit with BD Liquid Counting Beads (cat. # 349480, Becton, Dickinson and Company, BD Biosciences, San Jose, CA, USA) was used. According to the protocol, cells were stained with provided dyes, and cytometric analysis was conducted using the following parameters: fluidic flow rate 14 µL/min, the threshold set at 10,000 on (Forward Scatter-Height), sample volume set at 10 µL. The bacterial cells and counting beads were gated based on (Side Scatter) parameters and FL2, while the populations of alive, injured, and dead bacteria were discriminated based on an FL1 (thiazole orange) vs. FL3 (propidium iodide) plot. In live cells, the membrane is intact and impermeable to dyes, such as propidium iodide (PI), while when cells are injured or dead, the propidium iodide can leak into the cells because of their compromised membranes. PI is a nucleic acid intercalator, so it stains nucleic acids. On the other side, thiazole orange is a permeant dye that also reacts with nucleic acids but enters all cells—alive, injured, and dead, to varying degrees. Therefore, it will stain all cells containing nucleic acids. Thus a combination of these two dyes provides a rapid and reliable method for discriminating live, injured, and dead bacteria. To determine the concentrations of cell populations (expressed as cell/mL), Equation (1) was used:(1)$$\frac{\#\;of\;events\;in\;cell\;region}{\#\;of\;events\;in\;bead\;region}\times \frac{\#\;of\;beads\;per\;test{\;}^{*}}{test\;volume}\times dilution\;factor=concentration\;of\;cell\;population$$

* This value was found on the vial of BD Liquid Counting Beads and could vary from lot to lot.

In the case of *Lactobacillus plantarum*, changes in cell morphology under the influence of acrylamide were noted, which manifested in the form of cells with twice or several times stronger FL1 signal. Analysis of microscopic preparations stained by the Gram method confirmed that they were *Lactobacillus plantarum* cells appearing individually (bacillus), in pairs (diplobacillus), or in the form of chains (streptobacillus).

### 2.7. Statistical Analysis

All experiments were carried out in 5 replicates, and results are expressed as mean ± standard deviation (SD). When the impact of acrylamide on bacterial cell number was assessed, one-way analysis of variance (ANOVA) with Tukey’s honest significant difference (HSD) posthoc test was used to compare mean values and determine the significance of differences. The Brown–Forsythe test was used to verify the hypothesis of homogeneity of variances, while Shapiro–Wilk test was used to test the normality of distribution. A *p*-value < 0.05 was considered statistically significant. This part of statistical analysis was carried out in Dell Statistica (Data Analysis Software System, version 13, 2016, software.dell.com). Two-way ANOVA in a mixed model was used to assess data from flow cytometry, which means the interrelationship of two independent variables (incubation times and acrylamide concentrations) with a dependent variable (% of particular cell types), using IBM SPSS Statistics for Windows (2017, Version 25.0; IBM Corp., Armonk, NY, USA). Bonferroni posthoc test was used, and the differences were considered significant when *p*-value < 0.05. 

## 3. Results

### 3.1. Impact of Acrylamide on LAB Growth on Solid Medium 

The model medium used for experiments was low nitrogen and low carbon; therefore, the growth of *Lactobacillus* was significantly limited compared to the MRS medium. Acrylamide added to such medium did not show bactericidal or bacteriostatic activity against tested bacteria from the *Lactobacillus* genus even in very big concentrations, not reported in food (Table 1). 

Moreover, it was surprising that AA could stimulate the growth of *L. plantarum* and probiotic strain *L. acidophilus* LA-5 at a concentration of 1000 µg/mL, while *L. brevis* and *L. lactis* sp. *lactis* growth was more intense compared to control in the presence of 500 µg and 1000 µg of acrylamide per mL. The acrylamide concentration used in that part of the study was much higher than that detected in food. The concentrations of acrylamide reported in the literature vary from <10 to even 80,920 µg/kg, with the highest levels in potato chips, French fries, roasted coffee, and coffee extract [44,47,48,49,50]. Considering the quantities of particular foodstuffs we consume each day, it has been estimated that total AA uptake varies from 0.3 to 1.4 μg per kg body weight per day [48], depending on the age group (high consumption of coffee in adults) and eating habits. In particular cases, it can reach up to 5 μg/kg/day [5].

The obtained results suggested that some lactic acid bacteria probably could utilize acrylamide as a source of carbon and nitrogen if they lack in the environment (medium). The possibility of acrylamide degradation (not binding) by LAB has been suggested by the results of a study conducted on rats fed with acrylamide 3 h after consumption of four species of *Bifidobacterium*. A significant reduction in the degree of liver damage [51] has been observed. Other studies have demonstrated that in portions of potatoes prepared for French fries subjected to 15 min of fermentation before frying, the AA level is reduced by 90% [52]. However, these results only confirm that lactic acid bacteria can utilize substances that are precursors of acrylamide for their own use; the possibility of AA degradation by LAB has not been studied. 

Another strategy to reduce acrylamide formation in bread was proposed by Nachi et al. [53] by using selected lactic acid bacteria strains for dough fermentation. When the LAB was used to inoculate sourdough, the acrylamide concentration in the bread was reduced. This was due to the lower pH of the LAB-inoculated sourdough after fermentation for 16 h compared to the spontaneous sourdough (using only baker’s yeast). The acidification was accompanied by a significant increase in the concentration of reducing sugars, which were then used as electron acceptors by LAB and reduced to mannitol. The lack of sugar and low pH prevented the Maillard reaction. The most pronounced reduction of acrylamide formation (by 84.7%) was obtained in bread made with *Pediococcus acidilactici* strain S16.

### 3.2. Impact of AA on Lactic Acid Bacteria Concentration in Medium

Three LAB strains were chosen for further experiments: *L. brevis, L. plantarum,* and probiotic strain *L. acidophilus* LA-5. The bacteria concentration was measured by flow cytometry immediately and after 24 h and 48 h of acrylamide addition at various concentrations (Figure 1). The number of bacteria cells in the medium was determined according to Equation (1) using the number of events in the bacteria and bead regions and expressed as cell number per 1 mL.

Already at 0 h, some influence of AA on the number of bacteria in the limiting medium could be seen. It should be recalled that preparing cells for cytometric analysis takes about 2 h from the addition of AA to the medium, so the bacteria have time to change their metabolism and can already start using AA as a source of carbon or nitrogen. The presence of AA in the medium resulted in a decrease in *L. brevis* number after 24 h and 48 h incubation, but not significantly correlated with the AA concentration used (Figure 1B). For *L. acidophilus* LA-5, differences in population size compared to controls and between AA doses were not statistically significant (Figure 1A). The initial increase in cell number in the sample with 7.5 µg AA might have resulted from the use of small amounts of AA, but the concentration was too low to guarantee adequate conditions for bacterial growth and multiplication over a longer period of time. After 24 h, very large fluctuations in culture were reported, but after 48 h, the observed differences were not statistically significant, except for incubation in the presence of 15 µg/mL AA, when LA-5 was lower than in other samples.

These results suggested that *L. acidophilus* LA-5 was sensitive to AA because of decreased cell numbers, which would be consistent with the results of studies on other bacteria. However, these studies have used a medium-low in nitrogen and carbon, and not, as in most other studies, an optimal medium for LAB growth. Therefore, the LAB number also decreased in the medium without the addition of AA. This environment is, therefore, ideal for assessing the impact of AA in the absence of other, more absorbable sources of carbon and nitrogen. In milk, lactic acid bacteria use casein as a source of amino acids, thanks to having appropriate proteolytic enzymes [54]. Gene encoding the cell-wall bound proteinase (PrtP) is only found on the chromosome of *L. acidophilus*; neither *L. plantarum* nor *L. brevis* [55] has it. Also, some peptidases are unique to individual species. The presence of these enzymes, however, is important primarily in the environment typical for these microorganisms (milk) and affects the rate of multiplication of individual bacteria due to various assimilation possibilities of proteins available in the environment, as well as the final effect of the fermentation process, including the resulting secondary metabolites. 

In the medium used in this experiment, the only source of carbon and nitrogen was 0.45% of peptone obtained as enzymatic meat tissue hydrolysate, while, in MRS, usually about 2.5% of nitrogen compounds and 2% glucose are present. Furthermore, *L. acidophilus* and *L. brevis* possess all three known LAB peptide transport systems: the di/tripeptide Dpp and DtpT systems and the oligopeptide Opp system [55]. LAB are auxotrophs relative to amino acids, and, depending on the species, they can synthesize only a few amino acids, while the others must be provided with the medium. This means that a medium-low in protein and amino acids will quickly become a factor that limits bacterial growth because bacterial growth and multiplication require the assimilation of substrates to supply the cell with the necessary energy, carbon, and nitrogen to build new structures. That is why the ability to degrade acrylamide to be used as a source of nitrogen (released by NH_4_^+^ amidase) and carbon was so important in this experiment.

It should be recalled that in this part of the study, all cells were counted: those that were alive and able to function properly and further divide, those that were damaged and whose metabolism was temporarily switched to repair the damage, and dead cells that had not yet broken down.

The situation differed in the case of *L. plantarum* (Figure 1C). After 24 h in the control medium without AA, an increase in the number of bacteria was observed; however, by analyzing their morphology, it was clear that this correlated with the grouping type in which these cells occurred. In the initial population (0 h), 92.55 ± 0.08% of cells appeared as single bacilli and 7.41 ± 0.08% as diplobacilli, while no streptobacilli were observed. After 24 h, the number of cells in culture without AA increased slightly; however, this was mainly due to the fact that the diplobacilli split into single cells. A lack of division meant that after another 24 h, 99% of the population were single rods, and their numbers significantly decreased compared to the initial value.

### 3.3. Acrylamide Impact on LAB Viability

Cell concentration and viability were measured by flow cytometry immediately and 24 h and 48 h after acrylamide addition at various concentrations. Populations of dead, injured, and alive bacteria were discriminated based on fluorescence signal after staining with thiazole orange (FL1) and propidium iodide (FL2) provided in the assay. The concentrations of various cell populations were determined using counting beads. 

First, it was checked whether incubation time, regardless of acrylamide concentration, had a significant impact on the percentages of specific cell types (main effect of incubation time). All tested main effects of incubation time were statistically significant, except for dead cells of *Lactobacillus brevis* (Table 2). The viability of *L. brevis* increased after AA addition, while the percentage of injured cells was significantly diminished. Contrary to that was the viability of *Lactobacillus acidophilus* LA-5. The highest percentage of live cells was at the beginning of the experiment, while the lowest was after 24 h. Tracking changes in the percentage of injured cells, it appeared that some were repaired, and after 48 h, they were considered to be fully viable. Moreover, some dead cells underwent autolysis, and the cell content released in the medium was utilized by survivors. Lactic acid bacteria are characterized by differentiated autolytic activity, but the process allows them to eliminate weak or impaired cells from the population [56]. In the mentioned study, *L. plantarum* strains were autolyzed more than other LAB strains; however, the authors did not test the autolytic activity of *L. brevis* or *L. acidophilus* [56]. It is well known that some lactic acid bacteria can undergo enzymatic cleavage of cell wall peptidoglycans by peptidoglycan hydrolases present in the bacterial cells and that the autolysis depends on factors, such as carbon source, temperature, osmotic concentration, and pH [56]. It has also been demonstrated that N-acetylmuramidase has a critical function in *Lactobacillus bulgaricus* autolysis [57] as one of the major degraders of the cell wall.

In our experiment, the percentage of live *L. plantarum* cells was significantly lower after 48 h than at the beginning or after 24 h. Moreover, significant differences in *L. plantarum* morphology were observed. After 48 h of incubation in the presence of acrylamide, fewer cells were present in the form of single bacilli, while the amounts of diplobacilli and streptobacilli were significantly increased (Table 2 and Figure 2). Such an effect was not observed in other LAB strains.

Then, it was tested whether the acrylamide concentration, regardless of the time of incubation, had a significant effect on the percentage of specific cell types (alive, injured, dead) or *L. plantarum* morphology (main effect of acrylamide concentration). The ANOVA results are presented in Table 3 and posthoc tests in Table 4.

The analysis showed that the main effect of acrylamide concentration did not occur in the *L. brevis* strain, which meant that in this case, acrylamide (regardless of the incubation time) had no effect on their viability. The AA impact was observed in other tested strains and when the morphology of *L. plantarum* was taken into account. Posthoc analysis showed that acrylamide significantly increased the percentage of alive cells of *L. acidophilus* LA-5 strain, but this was only observed at a concentration of 30 μg/mL and was accompanied by a significant decrease in the number (percentage) of injured cells. In the *L. plantarum* strain, acrylamide at each concentration significantly reduced viability while also significantly increasing the number of injured cells. In addition, morphological examination of *L. plantarum* showed a decrease in the proportion of single cells (bacilli), mainly in favor of increasing their frequency in pairs (diplobacilli). The number of cells in the form of chains (streptobacillus) also increased, but to a lesser extent.

Finally, the interaction effects (simultaneous impact) of incubation time and acrylamide concentration were tested. Acrylamide significantly increased the viability of *L. acidophilus* LA-5 cells at a concentration of 30 μg/mL after 24 h incubation and at 0, 30, and 100 μg/mL after 48 h, when compared to the model medium without acrylamide (Table 5). The increase in alive cells was mainly accompanied by a reduction in injured cells, rather than dead ones. In turn, the viability of *L. acidophilus* LA-5 decreased at an acrylamide concentration of 15 μg/mL after 48 h.

In the case of *Lactobacillus brevis*, acrylamide at each concentration decreased the percentage of injured cells after 24 h and 48 h incubation (compared to control), although a statistically significant reduction in the percentage of alive bacteria was observed only after 48 h at 100 μg/mL (Table 6). All other changes were not statistically significant.

After 24 h and 48 h of incubation in the presence of acrylamide at concentrations higher than 7.5 μg/mL, reduced viability of *L. plantarum* cells was observed, while the number of injured cells increased compared with medium without acrylamide (Table 7).

We observed that the morphology of one of the tested bacteria, *Lactobacillus plantarum,* was significantly influenced by acrylamide. Based on the fact that cells with both twofold and several times stronger FL1 fluorescence signal (thiazole orange) appeared in the population, we concluded that acrylamide did not inhibit or even stimulate the division of *L. plantarum* but blocked cell separation; hence bacteria in the form of diplobacilli and streptobacilli were present in the population. This conclusion was confirmed by microscopic preparations. A statistically significant reduction in the number of single rods (bacilli) in the presence of AA in amounts of 7.5 and 15 μg/mL after 24 h incubation and in all AA concentrations after 48 h was demonstrated compared to the medium without acrylamide. This was mainly accompanied by a significant increase in the number of cells found in pairs (diplobacilli) and to a much lower extent in chains (streptobacilli). For each analyzed concentration of AA, this increase was especially significant after 48 h, reaching even ~50% at 30 µg/mL (Table 8). 

## 4. Discussion

In this study, we demonstrated that the tested lactic acid bacteria strains were tolerant of acrylamide even at high concentrations (up to 1 g/mL). Moreover, the growth of *Lactobacillus plantarum*, *L. lactis* sp. *Lactis,* and *L. brevis,* as well as probiotic strain *L. acidophilus* LA-5, was more intense in the presence of acrylamide at high concentration than in medium with limited accessibility of carbon and nitrogen compounds. The obtained results suggested that: (1) acrylamide had no toxic impact on LAB; (2) some lactic acid bacteria probably could utilize acrylamide as a source of carbon and nitrogen if they lack in the environment/medium. Of course, fermented milk beverages and the human gut cannot be considered nutrient-poor environments, as the availability of easily digestible food for bacteria is large, but the possibility of using acrylamide by lactic acid bacteria might be beneficial for both bacteria and the human intestine where the LAB reside. 

Our results proved that acrylamide not only influenced the number of lactic acid bacteria but also their viability. The impact of acrylamide on LAB viability depended on both the AA concentration and the bacteria species. First of all, when the impact of incubation time on bacterial viability was analyzed, all the main effects were statistically significant, except the percentage of dead cells of *Lactobacillus brevis*. Secondly, the main effect of acrylamide concentration on the percentage of alive, injured, and dead cells was not observed only in *L. brevis*. This suggested that *L. brevis* was less sensitive to acrylamide among the tested bacteria strains, and it was confirmed in the further analysis as almost all observed differences were not statistically significant. 

The posthoc tests showed that acrylamide caused a significant increase in the percentage of alive cells of probiotic strain *L. acidophilus* LA-5 at an AA concentration of 30 μg/mL compared to the cultures without AA. This increase was mainly accompanied by a reduction in the number of injured cells rather than dead ones. On the other side, acrylamide reduced the viability of *L. plantarum* cells after 24 h and 48 h incubation at each AA concentration except 7.5 μg/mL, simultaneously increasing the amount of injured cells. Moreover, we observed a strong influence of acrylamide (especially at a concentration of 30 µg/mL) on the morphology of bacteria only in *L. plantarum*. Based on the fact that cells with both twofold and several times stronger FL1 fluorescence signal (thiazole orange) appeared in the population, we concluded that acrylamide had no impact on the division of *L. plantarum,* but at the same time, it inhibited cell separation, as cells in the form of diplobacilli and streptobacilli were present in the population (confirmed in microscopic preparations). This suggested that in this case, acrylamide could have a harmful or even mutagenic impact on *L. plantarum*. 

It is known that many proteins and hydrolytic enzymes are involved in the proper growth and division of bacteria. Various enzymes participate in turnover (remodeling) of peptidoglycan, and their proper activity and specificity are critical, as bacterial division requires both localized hydrolysis and de novo biosynthesis of the peptidoglycan layer. For example, amidase and glucosaminidase displaying murein hydrolase activity are necessary for the generation of the equatorial ring on the staphylococcal cell surface and complete cell division and separation [58]. *Escherichia coli* division requires the activity of amidases—AmiA, AmiB, and AmiC [59]. It is important that muralytic enzymes distinguish elements of peptidoglycan of specific species. Generally, these enzymes are secreted into the surrounding medium, so they need to distinguish between the cell walls of other species and their own. It seems likely that the targeting mechanisms of murein hydrolases employ species-specific receptors for either physiological cell-wall turnover or the bacteriolytic killing of competing microorganisms [58,60,61]. Most Gram-positive bacteria contain a structurally similar peptidoglycan layer [62]. Thus, targeting of muralytic enzymes cannot be achieved by simple enzyme-substrate interactions but requires specific surface receptors [63]. For example, choline within teichoic acid moieties serves as a receptor for the LytA enzyme of *Streptococcus pneumoniae* [64]. A mutant of *S. pneumoniae* showing complete deletion in the lytA gene coding for N-acetylmuramyl-L-alanine amidase has been isolated. It shows a normal growth rate, and the most remarkable biological consequences of the absence of amidase are the formation of short chains (six to eight cells) and the absence of lysis in the stationary phase of growth. In our study, *L. plantarum* morphology changed in the presence of acrylamide, and bacteria started to form diplobacilli and streptobacilli. It is possible that acrylamide reacted with the active site of muralytic amidases and, therefore, blocked cell separation during division.

Different influence of AA on *Lactobacillus* species tested in the study could also be caused by the diversity of their teichoic acid (TA) structure. Teichoic acids in lactic acid bacteria consist of poly(ribitol phosphate) polymers with attached glucose, D-alanine, and/or glycerol molecules, among others [43]. Their structure is highly variable; thus, even closely related strains can differ in their ability to bind toxins. This is coincident with our results, showing that *L. brevis* was less and *L. plantarum* most sensitive to acrylamide among tested LAB strains. Serrano-Nino et al. [43] proved a significant correlation between the binding percentage of acrylamide and the content of some constituents of cell wall TAs. They proposed that H-bonds could occur between the carbonyl oxygen and the amino group (NH ··· OC) between adjacent acrylamide and D-alanine attached to the ribitol. Moreover, the amine group of D-alanine might react with acrylamide units by means of a Michael addition, while hydrogen bonds might also occur between carbonyl (C=O) oxygens of acrylamide and the hydroxyl groups of glucose residue or glycerol phosphate substituents attached to the poly (ribitol phosphate) chain. Moreover, they demonstrated that acrylamide binding to teichoic acids in *Lactobacillus* was irreversible.

The role of teichoic acids in cell division and morphogenesis has been investigated in some bacteria species, and it appears that wall teichoic acids (WTAs) are involved in elongation of bacteria, while lipoteichoic acids (LTAs) participate in the cellular division [65]. By obtaining the mutants of *L. plantarum,* it has been revealed that WTAs are not essential for survival, but they are required for proper cell elongation and cell division [66]. Therefore, the reaction of acrylamide with teichoic acids could impede division and cause that *L. plantarum* remains in the form of chains and diplobacillus.

Studies of Zhang [67] showed that the ability of acrylamide binding also depended on the peptidoglycan structure. The peptidoglycan of *L. plantarum* (strain 1.0065) had the highest affinity for AA binding (87.14%), whereas peptidoglycans of *L. casei* ATCC393 and *L. acidophilus* KLDS1.0307 showed lower affinity (75.50% and 56.75%, respectively). This binding ability of *L. plantarum* positively correlated with the carbohydrate content in peptidoglycan and the contents of four amino acids (alanine, aspartic acid, glutamic acid, and lysine). Additionally, it was demonstrated that C–O (carboxyl, polysaccharides, and arene), C=O amide, and N–H amines groups were involved in the AA binding. 

Analyzing the interaction of acrylamide with peptidoglycan, one should take into account the differences in the structure of cell wall stem peptides. The amino acid sequence of stem peptide involved in linking glycan chains in LAB peptidoglycan is L-Ala–D-Glu–X–D-Ala. The third amino acid (X) is a diamino acid, which in LAB usually is L-Lys (e.g., *L. lactis* and most lactobacilli), but can also be meso-diaminopimelic acid (mDAP) (e.g., in *L. plantarum*) or L-ornithine (e.g., in *L. fermentum*) [62]. Peptidoglycan with mDAP is typical for Gram-negative bacteria, and in such cell walls, a direct cross-connection between neighboring stem peptides takes place (the mDAP in position 3 of one peptide chain binds to D-Ala in position 4 of another chain). In lactic acid bacteria with Lys-type peptidoglycan, an additional interpeptide bridge made of one D-amino acid (e.g., D-Asp or D-Asn in *L. lactis*, *L. casei,* and most lactobacilli) is included [62]. It means that the structure of *L. plantarum* is unusual among LAB peptidoglycans, and it is different from the structure of other tested species. Additionally, this bacterium is characterized by a unique process among bacteria—O-acetylation of peptidoglycan [66]—which has an impact on *L. plantarum* autolysis. O-acetylation of N-acetylglucosamine (GlcNAc) inhibits the N-acetylglucosaminidase Acm2 (which is required for the ultimate step of cell separation of daughter cells), while O-acetylation of N-acetylmuramic acid (MurNAc) has been shown to activate autolysis through the activity of the N-acetylmuramoyl-L-alanine amidase LytH [68]. It is possible that acrylamide interacts with the mentioned enzymes (amidases) and hence influences cell division and separation. In our study, we observed that in the presence of AA, the *L. plantarum* morphology was changed, i.e., the percentage of cells in pairs or chains increased. It is worth mentioning that in *L. plantarum,* almost all the mDAP side chains are amidated. Defects of mDAP amidation in the *L. plantarum* mutant strain strongly affect the growth and cell morphology, causing filamentation and long-chain formation, suggesting that mDAP amidation may play a critical role in controlling the septation process [69]. Further studies are needed to explain whether acrylamide interacts with the amidation of mDAP or the activity of muralytic amidases. It is also possible that the presence of AA in low-carbon and low-nitrogen medium induces the synthesis of other amidases necessary for acrylamide degradation to acrylic acid and ammonia, but also able to cleave the amide bound in mDAP, influencing cell morphology.

The impact of acrylamide on LAB morphology should also be discussed in terms of the importance of bacterial aggregation on their functioning. First of all, bacterial aggregation (auto-aggregation) may facilitate biofilm formation by favoring bacterial attachment to surfaces or other microbes (co-aggregation). It also implicates better survival of LAB in the gut. Some studies indicate that biofilms are a stable point in a biological cycle that includes initiation, maturation, maintenance, and dissolution. According to O’Toole et al. [70], microbe development involves changes in form and function that play prominent roles in the life cycle of the organism, and biofilm formation is a prominent part of the lifestyle of microbes. Moreover, bacteria seem to initiate the development of biofilm in response to specific environmental conditions, such as nutrient availability. It has been proposed that the starvation response pathway can be subsumed as a part of the overall biofilm development cycle [70]. Secondly, when growing in biofilm, organisms become more resistant to higher deliverable levels of antibiotics or other antimicrobial compounds compared to single “suspended” cells [71]. The last matter is that aggregation and co-aggregation among bacteria are important in the prevention of colonization of surfaces by pathogens. It has been proved that some lactic acid bacteria are also able to control biofilm formation by pathogens and can, therefore, prevent the colonization of food-borne pathogens [72]. It is true, for example, for some *Lactobacillus plantarum* strains showing an aggregation phenotype [73].

## 5. Conclusions

In conclusion, we can assume that the tested strains of lactic acid bacteria found in the human digestive tract or in fermented milk drinks are tolerant to high concentrations of acrylamide (up to 1 g/mL). Some show better growth in medium with AA than in medium with limited carbon and nitrogen sources, suggesting the possibility that they use AA for their own metabolism. Of course, in the digestive tract, especially in the initial sections of the intestine, there is sufficient availability of easily digestible food, but the possibility of using AA is beneficial for both the lactic acid bacteria and the human in whose intestine the LAB resides.

Moreover, we can assume that eating AA-containing products with a properly functioning microbiota will be less harmful to human organs than previously thought. It is also good information for producers of food (e.g., yogurt) with the addition of AA-containing ingredients, such as roasted coffee, almond or nuts, muesli, baked biscuits, or cornflakes because it should not negatively affect the microorganisms necessary for their production.

## Figures and Tables

**Figure 1 nutrients-12-01157-f001:**
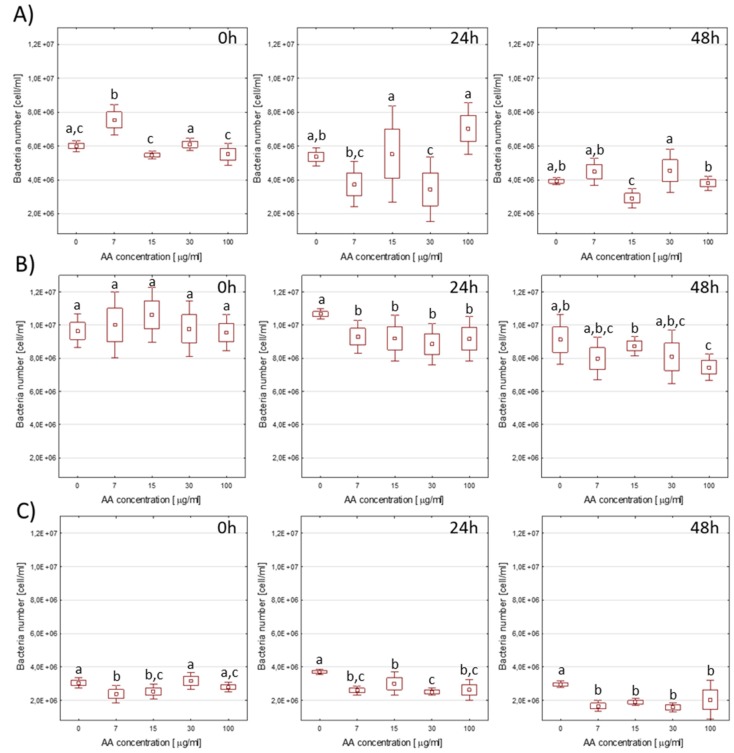
Impact of acrylamide (AA) concentration in medium (0, 7.5, 15, 30, and 100 µg/mL) on bacterial cell number in culture (cells/mL) determined by cytometric method immediately after AA addition and after 24 h and 48 h incubation. (**A**) *L. acidophilus LA-5,* (**B**) *L. brevis,* (**C**) *L. plantarum*. Values in graphs with different letters differ from each other at the level of *p* < 0.05 (Tukey’s HSD test).

**Figure 2 nutrients-12-01157-f002:**
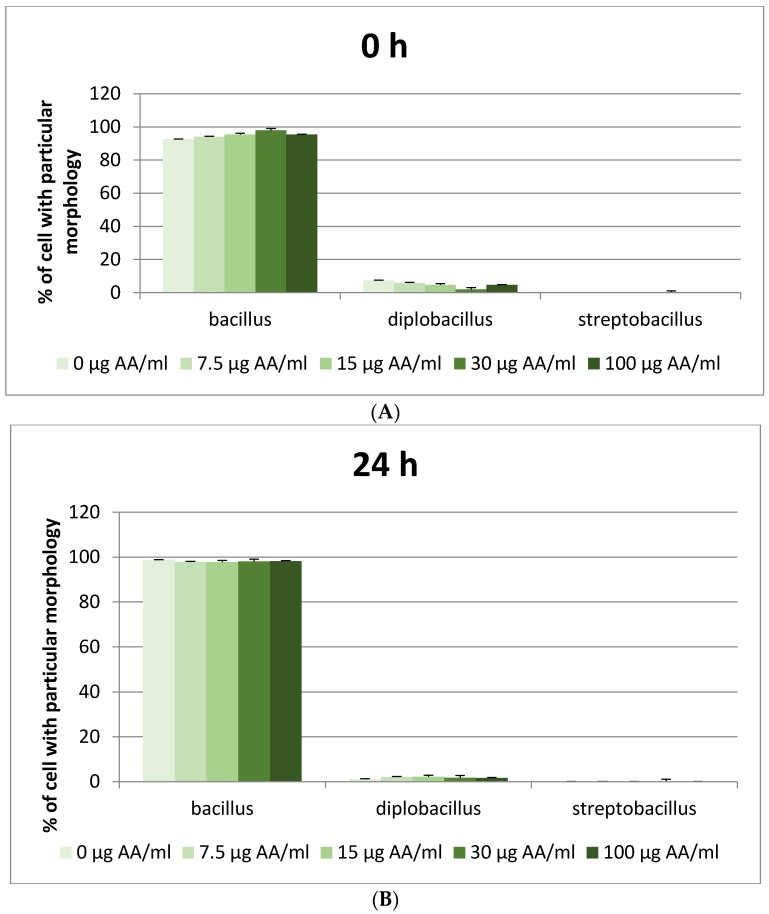
Impact of acrylamide concentration on the morphology of *Lactobacillus plantarum*. (**A**) 0 h, (**B**) 24 h, (**C**) 48 h.

**Table 1 nutrients-12-01157-t001:** Impact of acrylamide on the growth of *Lactobacillus* strains on solid medium.

Bacteria Strain	AA Concentration (μg/mL)	Growth Evaluation (Mean of 5 Replicates)
***Lactobacillus plantarum***	Control	++
	10	++
	50	++
	100	++
	250	++
	500	++
	1000	+++
***Lactobacillus brevis***	Control	++
	10	++
	50	++
	100	++
	250	++
	500	+++
	1000	+++
***Lactobacillus lactis* sp. *lactis***	Control	++
	10	++
	50	++
	100	++
	250	++
	500	+++
	1000	+++
***Lactobacillus casei***	Control	++
	10	++
	50	++
	100	++
	250	++
	500	++
	1000	++
***Lactobacillus acidophilus* LA-5**	Control	++
	10	++
	50	++
	100	++
	250	++
	500	++
	1000	+++
***Lactobacillus casei* LC-01**	Control	++
	10	++
	50	++
	100	++
	250	++
	500	++
	1000	++

Scale: ++++ very intense growth (colonies cover whole surface creating lawn plates); +++ intense growth (too many colonies to count, but they are distinguishable); ++ good growth (30–300 colonies/plate); + only a few colonies (<30 colonies/plate); – no growth. AA, acrylamide.

**Table 2 nutrients-12-01157-t002:** The main effect of incubation time: impact of time on percentage of specific cell types, regardless of AA concentration.

Bacteria Strain	% of Cells	Time			*F*	*p*
		0 h	24 h	48 h		
*Lactobacillus acidophilus LA-5*	alive	99.6 ± 0.35 ^a^	91.92 ± 2.08 ^b^	93.04 ± 3.43 ^c^	761.35	<0.001
	injured	0.39 ± 0.35 ^a^	6.60 ± 1.85 ^b^	5.34 ± 2.57 ^c^	937.62	<0.001
	dead	0.02 ± 0.02 ^a^	1.48 ± 0.48 ^b^	1.63 ± 0.93 ^b^	183.83	<0.001
*Lactobacillus brevis*	alive	98.76 ± 0.27 ^a^	99.51 ± 0.25 ^b^	99.54 ± 0.10 ^b^	107.44	<0.001
	injured	0.89 ± 0.26 ^a^	0.16 ± 0.06 ^b^	0.14 ± 0.03 ^b^	227.62	<0.001
	dead	0.35 ± 0.03	0.33 ± 0.20	0.32 ± 0.08	0.47	0.630
*Lactobacillus plantarum*	alive	97.42 ± 0.51 ^a^	97.62 ± 0.89 ^a^	95.09 ± 1.23 ^b^	143.33	<0.001
	injured	1.00 ± 0.17 ^a^	1.20 ± 0.36 ^b^	3.70 ± 1.33 ^c^	851.52	<0.001
	dead	1.58 ± 0.43 ^a^	1.18 ± 0.62 ^b^	1.21 ± 0.61 ^b^	7.59	0.002
Morphology *Lactobacillus plantarum*	bacillus	95.09 ± 2.35 ^a^	98.15 ± 0.50 ^b^	73.46 ± 15.96 ^c^	3358.82	<0.001
	diplobacillus	4.91 ± 2.35 ^a^	1.77 ± 0.50 ^b^	24.03 ± 13.47 ^c^	2909.62	<0.001
	streptobacillus	0 ± 0.01 ^a^	0.08 ± 0.02 ^b^	2.52 ± 3.18 ^c^	1078.97	<0.001

a, b, c—Means indicated with different letters differ from each other at the level of *p* < 0.05 (Bonferroni test). Values given are mean ± SD of the percentage of cells of a certain type.

**Table 3 nutrients-12-01157-t003:** The main effect of acrylamide concentration: influence of acrylamide concentration on the percentage of occurrence of certain cell types (expressed as arithmetic mean (SD)), regardless of the incubation time.

Bacteria Strain	% of Cells	Acrylamide Concentration (μg/mL)					*F*	*p*
		0	7.5	15	30	100		
*Lactobacillus acidophilus LA-5*	alive	93.86 (0.34) ^a^	94.79 (0.68) ^a^	93.48 (0.93) ^a^	97.18 (0.10) ^b^	94.95 (0.43) ^a^	31.67	<0.001
	injured	5.02 (0.27) ^a^	4.34 (0.54) ^a^	4.99 (0.57) ^a^	2.11 (0.03) ^b^	4.07 (0.44) ^b^	39.50	<0.001
	dead	1.12 (0.07) ^a^	0.86 (0.23) ^a^	1.54 (0.44) ^a^	0.71 (0.09) ^a^	0.97 (0.06) ^a^	9.62	<0.001
*Lactobacillus brevis*	alive	99.15 (0.02)	99.33 (0.04)	99.30 (0.16)	99.23 (0.19)	99.33 (0.04)	2.43	0.081
	injured	0.47 (0.01)	0.34 (0.03)	0.34 (0.03)	0.46 (0.18)	0.38 (0.01)	2.79	0.054
	dead	0.39 (0.01)	0.32 (0.03)	0.36 (0.13)	0.31 (0.02)	0.28 (0.04)	2.12	0.115
*Lactobacillus plantarum*	alive	97.81 (0.08) ^a^	96.76 (0.18) ^b^	95.84 (0.57) ^c^	96.51 (0.28) ^b^	96.64 (0.42) ^b^	19.95	<0.001
	injured	1.14 (0.04) ^a^	1.92 (0.09) ^b^	2.71 (0.17) ^c^	2.33 (0.13) ^d^	1.72 (0.19) ^b^	98.90	<0.001
	dead	1.05 (0.06) ^a^	1.32 (0.15) ^a^	1.45 (0.42) ^a^	1.16 (0.18) ^a^	1.64 (0.26) ^b^	4.50	0.009
Morphology *Lactobacillus plantarum*	bacillus	96.77 (1.03) ^a^	86.73 (0.51) ^b^	90.17 (0.29) ^c^	82.25 (0.38) ^d^	88.57 (0.71) ^b^	341.41	<0.001
	diplobacillus	3.17 (1.03) ^a^	12.21 (0.39)	9.51 (0.28)	14.95 (0.25)	11.36 (0.7) ^a^	263.77	<0.001
	streptobacillus	0.06 (0.01) ^a^	1.06 (0.13) ^b^	0.33 (0.04) ^c^	2.80 (0.24) ^d^	0.08 (0.01) ^a^	420.58	<0.001

a, b, c—Means with different letters differ from each other at the level of *p* < 0.05 (Bonferroni test).

**Table 4 nutrients-12-01157-t004:** The main effect of acrylamide concentration: posthoc Bonferroni test.

Bacteria Strain	AA Concentration (μg/mL)	% of Cells *					
		Alive/Bacillus		Injured/Diplobacillus		Dead/Streptobacillus	
		Difference between Means	*p*	Difference between Means	*p*	Difference between Means	*p*
*Lactobacillus acidophilus LA-5*	0 vs. 7.5	−0.93	0.182	0.68	0.197	0.26	0.898
	0 vs. 15	0.38	1.000	0.03	1.000	−0.42	0.093
	0 vs. 30	−3.32	<0.001	2.91	<0.001	0.42	0.093
	0 vs. 100	−1.10	0.067	0.95	0.020	0.15	1.000
*Lactobacillus brevis*	0 vs. 7.5	−0.19	0.187	0.12	0.332	0.06	1.000
	0 vs. 15	−0.16	0.460	0.13	0.223	0.02	1.000
	0 vs. 30	−0.09	1.000	0.01	1.000	0.08	0.724
	0 vs. 100	−0.19	0.180	0.08	1.000	0.10	0.197
*Lactobacillus plantarum*	0 vs. 7.5	1.05	0.002	−0.78	<0.001	−0.27	1.000
	0 vs. 15	1.97	<0.001	−1.57	<0.001	−0.40	0.186
	0 vs. 30	1.30	<0.001	−1.19	<0.001	−0.11	1.000
	0 vs. 100	1.17	<0.001	−0.58	<0.001	−0.59	0.012
morphology *Lactobacillus plantarum*	0 vs. 7.5	10.04	<0.001	−9.05	<0.001	−1.00	<0.001
	0 vs. 15	6.60	<0.001	−6.34	<0.001	−0.27	0.034
	0 vs. 30	14.52	<0.001	−11.78	<0.001	−2.74	<0.001
	0 vs. 100	8.20	<0.001	−8.19	<0.001	−0.01	1.000

* In a morphological study, percentages of bacillus, diplobacillus, and streptobacillus cells were compared.

**Table 5 nutrients-12-01157-t005:** The interaction effect of incubation time and concentration of acrylamide on the viability of *Lactobacillus acidophilus* LA-5 (Bonferroni test).

Incubation Time (h)	AA Concentration (μg/mL)	% of Cells					
		Alive		Injured		Dead	
		Difference between Means	*p*	Difference between Means	*p*	Difference between Means	*p*
0	0 vs. 7.5	−0.10	<0.001	0.08	0.001	0.02	1.000
	0 vs. 15	0.01	1.000	−0.02	1.000	0.01	1.000
	0 vs. 30	−0.02	0.334	0.01	1.000	0.01	1.000
	0 vs. 100	0.83	<0.001	−0.83	<0.001	0	1.000
24	0 vs. 7.5	0.58	1.000	−0.38	1.000	−0.20	1.000
	0 vs. 15	−2.15	0.149	2.23	0.015	−0.08	1.000
	0 vs. 30	−3.89	0.001	3.83	<0.001	0.06	1.000
	0 vs. 100	0.02	1.000	0.27	1.000	−0.29	1.000
48	0 vs. 7.5	−3.28	<0.001	2.33	<0.001	0.96	0.003
	0 vs. 15	3.29	<0.001	−2.12	<0.001	−1.17	<0.001
	0 vs. 30	−6.06	<0.001	4.88	<0.001	1.18	<0.001
	0 vs. 100	−4.13	<0.001	3.40	<0.001	0.73	0.029

**Table 6 nutrients-12-01157-t006:** The interaction effect of incubation time and concentration of acrylamide on the viability of *Lactobacillus brevis* (Bonferroni test).

Incubation Time (h)	AA Concentration (μg/mL)	% of Cells					
		Alive		Injured		Dead	
		Difference between Means	*p*	Difference between Means	*p*	Difference between Means	*p*
0	0 vs. 7.5	−0.11	1.000	0.15	1.000	−0.04	0.365
	0 vs. 15	−0.20	1.000	0.21	1.000	−0.01	1.000
	0 vs. 30	0.19	1.000	−0.15	1.000	−0.04	0.561
	0 vs. 100	−0.06	1.000	0.06	1.000	0.00	1.000
24	0 vs. 7.5	−0.32	0.435	0.15	<0.001	0.16	1.000
	0 vs. 15	−0.17	1.000	0.12	0.001	0.05	1.000
	0 vs. 30	−0.29	0.633	0.12	0.001	0.17	1.000
	0 vs. 100	−0.33	0.401	0.13	<0.001	0.20	1.000
48	0 vs. 7.5	−0.13	0.352	0.06	0.004	0.06	1.000
	0 vs. 15	−0.10	1.000	0.06	0.001	0.03	1.000
	0 vs. 30	−0.15	0.121	0.05	0.015	0.10	0.540
	0 vs. 100	−0.18	0.046	0.06	0.004	0.12	0.277

**Table 7 nutrients-12-01157-t007:** The interaction effect of incubation time and concentration of acrylamide on the viability of *Lactobacillus plantarum* (Bonferroni test).

Incubation Time (h)	AA Concentration (μg/mL)	% of Cells					
		Alive		Injured		Dead	
		Difference between Means	*p*	Difference between Means	*p*	Difference between Means	*p*
0	0 vs. 7.5	1.01	0.009	−0.09	1.000	−0.92	0.002
	0 vs. 15	0.75	0.098	−0.21	0.373	−0.53	0.149
	0 vs. 30	0.44	1.000	−0.04	1.000	−0.39	0.648
	0 vs. 100	0.48	0.789	−0.26	0.115	−0.22	1.000
24	0 vs. 7.5	1.00	0.153	−0.55	0.001	−0.45	1.000
	0 vs. 15	2.08	<0.001	−0.88	<0.001	−1.20	0.007
	0 vs. 30	1.46	0.010	−0.77	<0.001	−0.69	0.311
	0 vs. 100	1.44	0.011	−0.37	0.049	−1.07	0.019
48	0 vs. 7.5	1.12	0.260	−1.70	<0.001	0.57	0.481
	0 vs. 15	3.07	<0.001	−3.61	<0.001	0.53	0.652
	0 vs. 30	2.00	0.004	−2.76	<0.001	0.76	0.115
	0 vs. 100	1.58	0.028	−1.10	0.001	−0.48	0.911

**Table 8 nutrients-12-01157-t008:** The interaction effect of incubation time and concentration of acrylamide on the morphology of *Lactobacillus plantarum* (Bonferroni test).

Incubation Time (h)	AA Concentration (μg/mL)	% of Cells					
		Bacillus		Diplobacillus		Streptobacillus	
		Difference between Means	*p*	Difference between Means	*p*	Difference between Means	*p*
0	0 vs. 7.5	−1.51	1.000	1.52	1.000	0	1.000
	0 vs. 15	−2.77	0.140	2.77	0.140	0	1.000
	0 vs. 30	−5.42	<0.001	5.42	<0.001	0	1.000
	0 vs. 100	−2.82	0.127	2.81	0.127	0	1.000
24	0 vs. 7.5	0.87	0.024	−0.86	0.023	−0.01	1.000
	0 vs. 15	0.98	0.008	−0.98	0.007	0	1.000
	0 vs. 30	0.59	0.280	−0.57	0.313	−0.02	0.417
	0 vs. 100	0.48	0.692	−0.45	0.791	−0.02	0.598
48	0 vs. 7.5	30.77	<0.001	−27.80	<0.001	−2.98	<0.001
	0 vs. 15	21.60	<0.001	−20.81	<0.001	−0.80	0.033
	0 vs. 30	48.37	<0.001	−40.18	<0.001	−8.19	<0.001
	0 vs. 100	26.95	<0.001	−26.93	<0.001	−0.01	1.000

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
