# Peer review of "Is Acrylamide as Harmful as We Think? A New Look at the Impact of Acrylamide on the Viability of Beneficial Intestinal Bacteria of the Genus Lactobacillus"

_nutrients, 2020, doi:10.3390/nu12041157_

Round 1

Reviewer 1 Report

In this manuscript, the authors investigated the impact of acrylamide on  Lactobacillus bacteria using culture method and flow cytometry. The authors observed concentration- and species-dependent effects of acrylamide. Specifically, the viability of Lactobacillus acidophilus was increased, while of Lactobacillus plantarum decreased. Also,  acrylamide influenced the morphology of L. plantarum, probably by blocking the cells' separation during division.

I have only minor comments.

It would be useful to show statistical differences in graphs in Figure 1. It will be much easier to understand than reading the tables.

The authors should explain better how they discriminated alive, dead, and injured cells. Specifically, what the two dyes stain.

Author Response

Reviewer 1: It would be useful to show statistical differences in graphs in Figure 1. It will be much easier to understand than reading the tables.

Answer: As you suggested we added marks of statistical differences in graphs presented by Figure 1 (Line 318 in revised manuscript). We hope that now graph is more legible. Moreover, below the Figure 1 we added adequate explanation: “The values in particular graph differing in the letter index differ from each other at the level of p <0.05 (Tukey's HSD test).”

Reviewer 1: The authors should explain better how they discriminated alive, dead, and injured cells. Specifically, what the two dyes stain.

Answer: Of course, we introduced an adequate explanation into chapter Materials and Methods in the manuscript. In line 211 in revised manuscript we added the sentences: ““In alive cells the membrane is intact and impermeable to dyes such as propidium iodide (PI), while when cells are injured or dead the propidium iodide can leak into cells because of their compromised membranes. PI is a nucleic acid intercalator so it stains nucleic acids. On the other side, the thiazole orange is a permeant dye which also reacts with nucleic acids but it enters all cells, alive, injured and dead, to varying degrees. Therefore it will stain all cells containing nucleic acids. Thus a combination of these two dyes provides a rapid and reliable method for discriminating live, injured and dead bacteria.””

Reviewer 2 Report

This paper describes effects of acrylamide on growth and viability of lactic acid bacterial strains. The claim that the ability of such bacteria to use acrylamide as a nutrient might help detoxify acrylamide in the human gut is probably not physiologically relevant, as acrylamide bioavailability was not taken into account. Gut bacteria including LAB are mostly in the ileum and colon, but dietary acrylamide is rapidly absorbed before it could reach the bacteria in the lower gut. It is hard to imagine other circumstances, such as in LAB containing food products in which the ability of LAB to use or bind acrylamide would be likely or of importance, because such foods (e.g., yogurt) are unlikely to also be sources of acrylamide.

The apparent effects of acrylamide on some LAB morphology needs a bit more explanation as to who or how the aggregation of bacteria is important to LAB function.

The abstract should present brief key results quantitatively.

Methods seem sound, statistics appropriate, and results were described accurately.

Author Response

Reviewer 2: This paper describes effects of acrylamide on growth and viability of lactic acid bacterial strains. The claim that the ability of such bacteria to use acrylamide as a nutrient might help detoxify acrylamide in the human gut is probably not physiologically relevant, as acrylamide bioavailability was not taken into account. Gut bacteria including LAB are mostly in the ileum and colon, but dietary acrylamide is rapidly absorbed before it could reach the bacteria in the lower gut.

Answer: Indeed, bioavailability of acrylamide has not been studied in the work. It is known from the literature that acrylamide bioavailability after oral administration is high and depends on the organism species. In miniature pigs 99% of acrylamide dose is absorbed, in rats up to 90%, and in dogs about 70%. In a human volunteer study, 34% of an oral dose appeared in the urine during the first 24 h [Dybing et. al 2005]. According to Kadry et al. [1999] acrylamide is rapidly absorbed, but the feces still contained approximately 10% of the administered dose. It means that acrylamide can be present in colon. Moreover, there are available studies suggesting that some kind of food matrices (or food components) can reduce the intestinal absorption of acrylamide causing that unmetabolized acrylamide reaches the colon. For example, high protein concentrations in the human diet may reduce acrylamide uptake [Schabacker 2004].

We added such explanation into manuscript (in line 116).

Reviewer 2: It is hard to imagine other circumstances, such as in LAB containing food products in which the ability of LAB to use or bind acrylamide would be likely or of importance, because such foods (e.g., yogurt) are unlikely to also be sources of acrylamide.

Answer: Thank you very much for your doubts and suggestion. It is known that biscuits, muesli, roasted almonds, nuts or seeds, dried fruit, breakfast cereals, bran flake cereals contain a lot of acrylamide (according to the literature data sucha as e.g. Cressey et al 2005, Sadowska-Rociek et al. 2018, Amrein 2005, even up to ~1500 µg/kg). In the markets, there is a lot of milk fermented products which contain mentioned above “additives”, but also so called “pro-healthy food” is now available for consumers. All those probiotic bars and probiotic cereals contain live probiotic strains of LAB together with crispy cereals, roasted nuts, almonds and seeds, almond or peanuts butter, dried fruits, flakes etc… So, it is not only possible, but it is already happening on the market that we have LAB and acrylamide in one food product.

We added explanations as above into revised manuscript (in line 110).

Reviewer 2: The apparent effects of acrylamide on some LAB morphology needs a bit more explanation as to who or how the aggregation of bacteria is important to LAB function.

Answer: Thank you for paying attention to another aspect of changes in L. plantarum morphology. We added more explanation in the manuscript (line 601 in Discussion).

First of all, bacteria aggregation (auto-aggregation) may facilitate biofilm formation. Aggregation may favor attaching of bacteria to a surface or other microbes (co-aggregation). It implicates better survival of LAB in gut. Some studies indicate that biofilms are a stable point in a biological cycle that includes initiation, maturation, maintenance, and dissolution. According to O’Toole [2000] microbes development involves changes in form and function that play a prominent role in the life cycle of the organism and biofilm formation is a prominent part of the lifestyle of microbes. Moreover, bacteria seem to initiate the development of biofilm formation in response to specific environmental conditions, e.g. nutrient availability. It has been proposed that the starvation response pathway can be subsumed as a part of the overall biofilm developmental cycle [O’Toole 2000].

Secondly, organisms when growing in biofilm become more resistant to the highest deliverable levels of antibiotics or other antimicrobial compounds comparing to single cells [Vergeres 1992]. And the other question is that aggregation and co-aggregation among bacteria play an important role in prevention of colonization of surfaces by pathogens. It has been proved that some lactic acid bacteria are also able to control biofilm formation by pathogens and can therefore prevent the colonization of food-borne pathogens [Gomez 2016] . It is true for example for some Lactobacillus plantarum strains showing an aggregation phenotype [García-Cayuela et al. 2014].

Reviewer 2: The abstract should present brief key results quantitatively.

Answer: According to the Journal requirements, the Abstract paragraph has limit of 200 words and should contain Background, Methods, Results, Conclusions. Hence, its structure does not allow for providing more details. In our opinion, presenting values only in one case, without providing them for the other, does not make sense. Therefore, we had to delete the quantitative description of the results because of exceeding the allowed word limit. For this reason, abstract contains only a general description of trends, not numerical values. 

Reviewer 2: Methods seem sound, statistics appropriate, and results were described accurately.

Answer: Thank you very much for yor review.

Round 2

Reviewer 2 Report

Appropriate revisions were included, responsive to reviewer comments.